# Automatic Recognition of Giant Panda Attributes from Their Vocalizations Based on Squeeze-and-Excitation Network

**DOI:** 10.3390/s22208015

**Published:** 2022-10-20

**Authors:** Qijun Zhao, Yanqiu Zhang, Rong Hou, Mengnan He, Peng Liu, Ping Xu, Zhihe Zhang, Peng Chen

**Affiliations:** 1National Key Laboratory of Fundamental Science on Synthetic Vision, Sichuan University, Chengdu 610065, China; 2College of Computer Science, Sichuan University, Chengdu 610065, China; 3Chengdu Research Base of Giant Panda Breeding, Sichuan Key Laboratory of Conservation Biology for Endangered Wildlife, Chengdu 610086, China; 4Sichuan Academy of Giant Panda, Chengdu 610086, China; 5Giant Panda National Park Chengdu Administration, Chengdu 610086, China

**Keywords:** giant panda, attribute recognition, bioacoustics, species conservation, deep learning, SENet

## Abstract

The giant panda (*Ailuropoda melanoleuca*) has long attracted the attention of conservationists as a flagship and umbrella species. Collecting attribute information on the age structure and sex ratio of the wild giant panda populations can support our understanding of their status and the design of more effective conservation schemes. In view of the shortcomings of traditional methods, which cannot automatically recognize the age and sex of giant pandas, we designed a SENet (Squeeze-and-Excitation Network)-based model to automatically recognize the attributes of giant pandas from their vocalizations. We focused on the recognition of age groups (juvenile and adult) and sex of giant pandas. The reason for using vocalizations is that among the modes of animal communication, sound has the advantages of long transmission distances, strong penetrating power, and rich information. We collected a dataset of calls from 28 captive giant panda individuals, with a total duration of 1298.02 s of recordings. We used MFCC (Mel-frequency Cepstral Coefficients), which is an acoustic feature, as inputs for the SENet. Considering that small datasets are not conducive to convergence in the training process, we increased the size of the training data via SpecAugment. In addition, we used focal loss to reduce the impact of data imbalance. Our results showed that the F1 scores of our method for recognizing age group and sex reached 96.46% ± 5.71% and 85.85% ± 7.99%, respectively, demonstrating that the automatic recognition of giant panda attributes based on their vocalizations is feasible and effective. This more convenient, quick, timesaving, and laborsaving attribute recognition method can be used in the investigation of wild giant pandas in the future.

## 1. Introduction

The giant panda (*Ailuropoda melanoleuca*) is a unique and vulnerable animal endemic to China. Pandas are also a flagship and umbrella species for biodiversity conservation. This means that pandas can draw significant resources into conservation efforts and that working to conserve pandas will inevitably protect the other species that share their habitat [1]. Researchers have put major effort into investigating the population size and population trends of pandas in order to make appropriate conservation plans [2,3,4]. Age structure and sex ratio are two factors that influence the population dynamics of pandas [5,6]. Age structure refers to the proportion or configuration of different age groups within a population, and it has a substantial impact on the overall birth rate and mortality of a population. Research on the age structure of panda populations can help conservation managers monitor and predict the population dynamics of pandas in real-time [7]. The sex ratio of a population refers to the relative number of male and female individuals. In wild populations, changes in sex ratio have often led to changes in intraspecies relationships and mating behaviors, which in turn affects a population’s growth rate [8].

Traditionally, there have been two main methods for investigating panda’s age structure and sex ratios: bite-size and DNA-based approaches. The bite-size approach involves examining intact pieces of bamboo stems in panda feces. The age and sex information of pandas can be obtained by measuring the distribution of fragment lengths, which reflects the bite lengths of the animal [7,9]. This method only works for adult pandas because panda cubs eat bamboo leaves, not stems. The DNA-based approach was first utilized in the fourth decennial panda population survey. When fresh feces were found in the wild, researchers preserved the fecal samples in 100% ethanol. The panda’s intestinal cells in the outer mucosa of the feces were then isolated to extract DNA [10]. The DNA-based approach has very high requirements for panda feces, and only fresh feces that retain a mucous membrane can be used. Both methods demand large amounts of effort on the part of field and laboratory staff, which is expensive, inefficient, and potentially dangerous for field researchers.

Fortunately, deep learning offers new ways for computer power to replace or amplify the efforts of researchers. For example, giant panda images and deep learning can be used to identify the current behavior of pandas [11], while computer analysis of images of pandas has been shown to efficiently classify the age category and sex of pandas [1]. Unfortunately, this technique requires high quality images of pandas which are difficult and expensive to collect under in-the-wild conditions. Sound recordings, however, allow data to be collected across long transmission distances, since sound can travel farther and penetrate further than light, while still carrying rich information [12]. Pervious work has shown that both source and filter-related acoustic features of a type of giant panda vocalization referred to as a “bleat” provided reliable information on a caller’s sex, age, and body size [13]. Therefore, it is feasible to use the vocalizations of pandas to obtain their age and sex information.

### 1.1. Acoustic Recognition of Animals

Motivated by the need for better conservation tools, researchers have begun applying speech recognition algorithms to animal vocalizations [14]. Schröter et al. constructed a new model to recognize different types of killer whale calls [15]. The accuracy can reach 97%, and successfully visualized the feature map. Solomes and Stowell found it is possible to automatically identify species of birds according to features of their vocalizations [16], and results indicate a reliable accuracy of 87% for the classification of 12 call classes. Other researchers have used deep learning methods to automatically recognize different states of a beehive using audio as the input [17]. Kiskin et al. utilized a dataset of audio recordings of mosquitoes with 70.3% accuracy to classify species by their flight tones [18].

Investigators have also applied bioacoustics tools to panda calls, with one team proposing a method that achieved an accuracy of 88.33% on data from 20 individuals [19]. However, the technique the team used for the recognition of adult pandas was not an end-to-end automatic recognition system with the feature extraction independent of the network. Others have devised a system to automatically predict the mating success of giant pandas based on their calls during breeding encounters with an F1 score of 88.7% [20]. In the above two papers, the subjects were only adult pandas.

Our previous work filled in the shortcoming of existing recognition methods that do not use vocalizations from young pandas. The results proved that panda vocalizations can be used to recognize age groups by deep learning [21]. However, when training the network, the vocalizations from the training data and the test data sometimes came from the same panda. This method of data organization will improve the recognition result.

### 1.2. Content of This Study

This study expands on our previous work [21] from the following aspects. (i) Vocalizations from sub-adult pandas were included, whereas the dataset in [21] contains only adult pandas and cubs aged under 100 days. (ii) The data were carefully organized such that individuals do not overlap between training and testing data while there is no such requirement in [21]. (iii) We further showed that both age groups and sex of giant pandas can be automatically recognized from their vocalizations, which is implemented through the same network, while in [21] only the age groups were recognized.

## 2. Materials and Methods

### 2.1. Materials

#### 2.1.1. Dataset

The data used in this paper were acquired at the Chengdu Research Base of Giant Panda Breeding, Sichuan, China from 2019 to 2021, using Shure VP89m, TASCAM DR-100MKII and SONY PCM-D100. Recordings were made of 28 captive pandas in total. When a panda reaches sexual maturity, it is considered to be an adult. The pandas in the dataset belonged to four age groups: cub, sub-adult, adult, and geriatric. We choose the call data of adult and geriatric pandas from the breeding seasons and the calls of cubs and sub-adult pandas that were made while playing. Prior to training the recognizer, we binned these four age groups into two groups: ‘juvenile’, consisting of cubs and sub-adults, and ‘adults’, consisting of adult and old age groups. The sex labels were simply male and female. The individuals in the dataset included 8 juvenile females, 3 juvenile males, 13 adult females, and 4 adult males. When training, we ensured that the individual pandas in the training and the test sets are different to simulate a more realistic scenario.

#### 2.1.2. Data Preprocessing

The vocalizations were all recorded dual-channel, with varying sampling rates of 192,000 Hz, 48,000 Hz and 44,100 Hz. We converted all calls from dual-channel to single-channel and normalized the sampling rate of all recordings to 44,100 Hz. Reducing the sampling rate to 44,100 Hz can effectively reduce the time of data preprocessing while maintaining the compact disc (CD) quality of audio [22]. To ensure the consistency of data dimensions during training, we divided the original call clips of giant pandas into 2-second segments without overlap, and the segments whose duration were shorter than 2-second were expanded to 2-second via zero-padding on the log-mel spectrum. In audio signal processing, it is very common to pad the log-mel spectrum by filling zeros or copying a part of the log-mel spectrum to make its length consistent. However, compared with padding via copying, zero-padding can reduce errors in the log-mel spectrum [23]. The reason for the division into 2-seconds is that in the collected data, the duration of different types of vocalizations of giant pandas are almost all 2 seconds. In addition, we had to make a very strong assumption that there is a vocalization of only one individual panda in each clip which can reduce the difficulty of the task. We manually excluded call clips that contain multiple pandas or are contaminated by other sounds (e.g., human voice or ambient noises). Because of the conditions where the recordings were made, the call recordings of cubs had relatively weak background noise, while those of others generally had stronger background noises (e.g., the sound of working air-conditioners). In order to avoid training the recognizer on these background features of the recordings, we collected examples of only background sound appearing in the data of adult pandas and then integrated them into the calls of the panda cubs by mixing the recordings together. After these processing steps, the total duration of the call data was 1298.02 s (Figure 1).

#### 2.1.3. MFCC

The main frequency component of speech is called formant, which carries the identification attributes of a sound, just like a personal ID card [24]. In this paper, we employed the MFCC (Mel-frequency Cepstral Coefficients) feature as the input because it can contain formant information [25]. Peter and Zhao respectively described the call structure and call frequency of adult pandas and baby pandas [26,27]. We found that MFCC is equally suitable for extracting the acoustic features of panda’s vocalizations. As an acoustic feature, MFCC has strong anti-noise properties and is widely used in the field of speech recognition. To derive MFCC from original audio, we took a number of steps. First, the original audio needed to be divided into frames. The frame collects N sampling points into one observation unit, typically with a duration of ~20–30 ms. In order to avoid excessive changes in two adjacent frames, there will be an overlapping area between two adjacent frames. After framing, we multiplied each frame by a Hamming window to increase the continuity between the left and right ends of the frame. Fast Fourier transformations (FFT) were used to convert the signal from the time domain to the frequency domain so that we could better observe the signal characteristics. Therefore, after multiplying by a Hamming window, each frame must undergo a FFT to obtain the energy distribution on the frequency spectrum. We then passed the frequency spectrum through mel filters, before calculating the logarithmic energy of the output of the filter. Mel filters were the most important operation in this process, which could eliminate the effect of harmonics and highlight the formants of the original call data. Finally, the MFCC was obtained by the discrete cosine transform function (DCT).

### 2.2. Methods

We divided the vocalization data by individual to ensure that the vocalizations in the training dataset and test dataset come from different individuals. This approach reflected real-world considerations; we usually only get the vocalization data of certain individuals for training, but we hoped that the trained model was also effective for other individuals. The size of the dataset in this paper is small and very imbalanced, due to many more vocalizations being available from certain individuals or age and sexes. We attempted to mitigate this issue through several steps. After considering several approaches for dealing with small and imbalanced datasets, we opted to use data augmentation and focal loss to attempt to improve our experimental results (Figure 2). After we processed the vocalization data from the pandas, we began to train the neural net. In this process, we only utilized one neural network, named SENet [28], to recognize different attributes of the pandas. In previous work, we used two different types of networks with one focused on local features and the other on contextual [21]. We found that SENet, a network that pays more attention to local features, was more suitable for our task. We labeled training data with two tags, one for each attribute, yielding four groups (i.e., sub-adult female, sub-adult male, adult female, and adult male). After labeling the call data, we input the MFCC in SENet to get recognition results.

#### 2.2.1. Model Architecture

SENet [28] is a type of convolutional neural network (CNN) [29] that pays attention to the relationship between channels. SENet utilizes a Squeeze-and-Excitation (SE) module, which attempts to let the model automatically learn the importance of different channel features. Squeeze operates on the feature map. This operation uses a global average pool on each channel of the feature map to get the scalar of each channel. Then excitation is performed on the results obtained by the squeeze, in order for the CNN to learn the relationship between each channel, and also infers weights of different channels. Finally, the process is completed by multiplying the result by the original feature map to get the final feature. The advantage of this model is that it can pay more attention to the channel features with the most information while suppressing those unimportant channel features.

#### 2.2.2. Data Augmentation

Considering that our dataset is small and that there is an obvious imbalance between the volume of vocalization data of four groupings of pandas, we augmented the call data by adding Gaussian noise and applying SpecAugment [30]. Gaussian noise refers to a type of noise whose probability density function obeys a Gaussian distribution. SpecAugment operates on the log-mel spectrum. Specifically, we applied frequency masking by adding a mask with a value of 0 to the frequency axis of the log-mel spectrum, and time masking by adding a mask with a value of 0 to the time axis (Figure 3). As needed, we could set the number of masks, the width of masks, and identify which part of the log-mel spectrum needs to be masked. Data augmentation reduces the impact of data imbalance by increasing the amount of data.

#### 2.2.3. Focal Loss

The dataset has large imbalances in both age and sex (Figure 2). In addition to using data augmentation, we utilized focal loss [31] to solve this problem. Focal loss improves on the cross entropy loss which uses an adjustment item in the cross entropy to focus the learning on hard examples. Focal loss in the form of multiple categories was performed following the procedures in [1]. $p_{t}$ is predicted probability. If a sample predicts well, then the loss generated by this sample is close to 0. The role of $\alpha_{t}$ is to solve the imbalance of positive and negative samples and the γ is mainly used to solve the imbalance of difficult and easy samples.
(1)$$FL\left(p_{t}\right)=-\alpha_{t}{\left(1-p_{t}\right)}^{\gamma }log\left(p_{t}\right)$$

### 2.3. Implementation Detail

The proposed method was implemented on Ubuntu with an NVIDIA GTX 1080. After processing the original vocalization data, the dimension of all data was (173, 64, 1). When we extracted MFCC, some parameter settings were important. They are ‘n_mels’, ‘n_fft’ and ‘hop_length’. The ‘n_mels’ means the number of mel filters, which was set to 64. ‘n_fft’ means the length of the FFT window which was 1024. ‘hop_length’ refers to the overlapping area which was 512. The network’s batch size was 32. The learning rate was $1\times 10^{-3}$ and the epoch was 100.

As in [21], we first carried out evaluation experiment without considering the mutual exclusion of individuals in training and testing sets. We refer to the experiment as Experiment 1. Ten-fold cross validation was conducted in the experiment.

To evaluate the effectiveness of the methods in more practical scenarios where individuals in testing are usually not seen in training, we conducted Experiment 2 in this paper by ensuring that individuals in training and testing sets are mutually exclusive. We completed three sub experiments with different setups for both sex recognition and age group recognition, including (A) no augmented training data nor use of cross entropy loss, (B) augmented training data and use of cross entropy loss, (C) augmented training data and use of focal loss. Sub-experiment 2A is used as a control with Experiment 1 to study the impact of controlling the data distribution. Sub-experiments 2B and 2C were conducted to evaluate techniques for solving the problems of small dataset size and data imbalance, to determine if they improved the recognizer.

The number of pandas and the amount of call data we could collect was very limited. We adopted the principle of “lower, not higher” for the training data. That means when training data and test data were from different individuals, the training data were minimal and the number of training data samples was fixed. To do this, we first chose four individuals randomly as test data. If the four individuals had less than 100 vocalization clips in total, we copied clips to increase the vocalization data to 100 samples. When the number of vocalization clips was more than 100, we randomly subsampled 100 of them. Among the remaining permutations of 24 individuals, 540 was the lowest number of vocalization clips, meaning that 540 vocalization clips were the input size for both sex and age group. In this way, we created ten training and test datasets for each sub-experiment.

The data augmentation in Experiments 2B and 2C were the same, including in the number methods of data augmentation. This allowed us to see the effectiveness of focal loss more intuitively. Before training, we augmented male data six times to meet the number of female data and augmented “juvenile” data twice, and “adults” once. This allowed to balance the data in the input network to make the data of each category more consistent.

In addition to the above experiments, we also evaluated the impact of the training data size on the recognition performance. We refer to this experiment as Experiment 3, in which we fixed the individuals in the test set and set the size of the test set to 100 as above. The size of the training set was increased from 540 by increments of 30 clips up to a maximum of 720. Experiment 3 utilized data augmentation and focal loss. Note that the individuals in training data and test data are mutually exclusive in Experiment 3. Table 1 summarizes the settings of different experiments in this paper, and Table 2 reports the training time required for each experiment.

## 3. Results and Discussions

### 3.1. Model Evaluation

Our dataset is very small, which means that it was better to use F1 scores as an evaluation method. The formula of F1 score follows Equation (2). Precision refers to the proportion of positive samples in the positive examples determined by the classifier as in Equation (3), and recall refers to the proportion of the total positive samples that are predicted to be positive as per Equation (4).
(2)$$F1=2\times \frac{precision\times recall}{precision+recall}$$
(3)$$precision=\frac{TP}{TP+FP}$$
(4)$$recall=\frac{TP}{TP+FN}$$
where *TP*, *FP*, and *FN* in (3) and (4) mean true positive, false positive, and false negative, respectively. TP means that the predicted label matched the true label (e.g., a clip of an adult was correctly identified as an adult). FP indicates that the predicted label was positive for an attribute characterization, while the true label was negative (e.g., if the male recognizer model classified a clip as containing a male panda, but the clip did not contain a male panda vocalization). Finally, *FN* means that the recognizer failed to identify the attribute of interest in a clip (e.g., the model classified a clip as not containing a juvenile panda when it in fact did).

In this paper, we compute the average and standard deviation of each of the metrics (i.e., precision, recall, and F1 scores) in cross-validation evaluation experiments. For a method, higher average metric values indicate that it has better accuracy, while smaller standard deviations mean that it has better stability (or better generalization ability).

### 3.2. Age Group Recognition Results

For the age group recognition, we wanted to know whether or not the vocalizing panda is an adult based on their vocalization characteristics. In Experiment 1, we let individuals appear in training data and test data at the same time, which means that individuals in the training data and test data were not mutually exclusive. We trained the model 10 times, each time setting the size of training data and test data to 540 clips and 100 clips, respectively. The average F1 score (±standard deviation) was 98.28% (±1.27%). We did not perform any data augmentation nor use focal loss when training, but it was a very impressive result.

In other sub-experiments, we re-organized the data so that the individuals in the training and test data were mutually exclusive. Sub-experiments 2A and 2B yielded F1 scores of 83.23% (±13.21%) and 88.86% (±7.14%), showing an improvement resulting from the use of data augmentation and cross entropy loss. In Experiment 2C, where cross entropy loss was replaced by focal loss, the F1 score was 96.46% (±5.71%) (Table 3).

The F1 score declined when vocalizations from individuals were divided between training and test sets no matter what technique was applied, but Experiments 2B and 2C showed that data augmentation and focal loss did help mitigate the challenges of data imbalance.

Experiment 3 showed the value of increasing the amount of training data. Our dataset is very small, so in previous experiments, we kept the size of the training data to the minimum 540 clips. Experiment 3 mainly showed that our model can further improve the F1 score when the amount of training data increases (Figure 4). Table 4 gives the quantitative results. Obviously, using more training data is beneficial to improving the performance of the recognition model. Specifically, as the size of training data increases, the average accuracy in terms of F1 scores is in general enhanced. Although in our experiments the F1 improvement becomes marginal as the number of training data reaches 630 or more, the standard deviation keeps being reduced. Collectively, the experiments showed that our model and methods can be further improved when the training data become larger.

### 3.3. Sex Recognition Results

For the sex recognition, we controlled for individual mutual exclusion in the training and test data in the same way. In this task, we simply wanted to know whether the panda is female or male by vocalization. The organization of the experiments was exactly the same as above for age group recognition. Experiment 1 had an average F1 score of 94.44% (±1.56%). For Experiments 2A and 2B, the F1 scores were 80.42% (±15.64%) and 81.77% (±10.96%), respectively, again depending on whether data augmentation and cross extropy loss were utilized. The F1 score of Experiment 2C was 85.85% (±7.99%), showing an increase in F1 score when focal loss replaced cross entropy loss. Table 5 shows the exact value.

The best method to solve this problem is collecting more data. Our findings can also help to apply this work in the real world. While the forest is a more complex acoustic environment, we hope to have more data for training, so that the model can ultimately make stronger generalizations. More data will include more individuals and more balanced samples.

Experiment 2C mainly showed that our model can further improve the F1 score when the amount of training data increases (Figure 5). According to the quantitative results in Table 6, a similar observation can be made. As the number of training data increases, the improvement on average F1 score gradually becomes slight and even oscillates, whereas the standard deviation remains reduced.

### 3.4. Discussions

We found that it is feasible to recognize attributes of pandas by the acoustic features of their vocalizations when they were transformed to MFCC. The distribution of attribute information contained in the vocalizations of different panda individuals was different. For both recognition tasks, using data augmentation and focal loss improved the F1 score and reduced the standard deviation.

It is worth noting that the standard deviation of sex recognition success was high, and the improvement of F1 score for sex was not as obvious as that for age groups, even after utilizing data augmentation and focal loss. We think that the distributions of females and males were very close, leading to the challenge of using limited audio recordings to assess the sex of a panda. Now, the data set we have can only cover very limited distributions, which led to very large standard deviations. The imbalance of female and male data was much larger than that of age group and just using focal loss could not completely solve the data imbalance.

## 4. Conclusions

As the first to study the potential of automatically recognizing giant panda attributes from their vocalizations, we collected a dataset of giant panda vocalizations including female and male, juvenile and adults. As a first, we were able to automatically predict panda age groups and sex via their vocalizations using the same neural network, SENet. Our results showed that even for a small and imbalanced dataset, the challenges of giant panda attribute recognition can be partly overcome. For each recognition task, we conducted the same three experiments. Specifically, our three experiments for both age and sex recognition showed that data augmentation and focal loss improved F1 scores for the recognizer. Experiment 3 showed that our model can be improved more when the amount of training data increased.

Yet, this study still has limitations. The collected data will inevitably have noise. Our study was predicated on the assumption that a call clip only had one panda. In this paper, we directly eliminated these noises, which reduced the size of our dataset. Besides, the data augmentation methods that we used in this paper do not consider the consistency of samples before and after data augmentation at all. This may impact our results. Nevertheless, as the first attempt to automatically recognize giant panda attributes from their vocalizations, this study demonstrates the potential of using vocalization data to investigate the age group and sex of giant pandas, which can significantly improve the giant panda population survey in terms of manpower, timeliness, and funding. It is possible to introduce deep learning into the conservation of wild giant pandas. Conducting research on giant panda populations based on vocalization analysis has high theoretical significance and practical application prospects for giant panda conservation, wildlife conservation in general, and research on computer-related technologies.

Currently, we are continuing to collect more data. For future work, we plan to reduce the impact of noise in the collected data via data preprocessing and learning techniques rather than via data selection. Furthermore, studying how to separate the calls of multiple pandas from a clip will also be an important and a necessary direction. We are also going to design a new data augmentation method to make the semantic information before and after data augmentation as similar as possible.

## 5. Patents

Chinese invention patent: A method and system of giant panda attributes recognition based on vocalizations [2022107915858] has been granted.

## Figures and Tables

**Figure 1 sensors-22-08015-f001:**
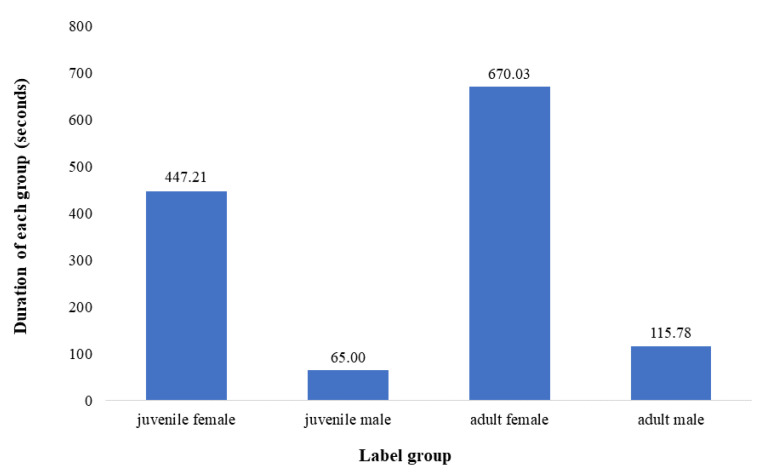
The total duration of call recordings from each grouping of pandas in this study: juvenile female, juvenile male, adult female, and adult male.

**Figure 2 sensors-22-08015-f002:**
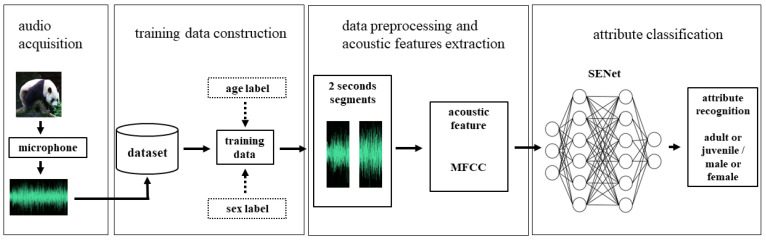
An illustration of the methods used in this study, from audio acquisition to final attribute classification. Note that age group classification and sex classification were done separately, and both were approached as binary classification tasks.

**Figure 3 sensors-22-08015-f003:**
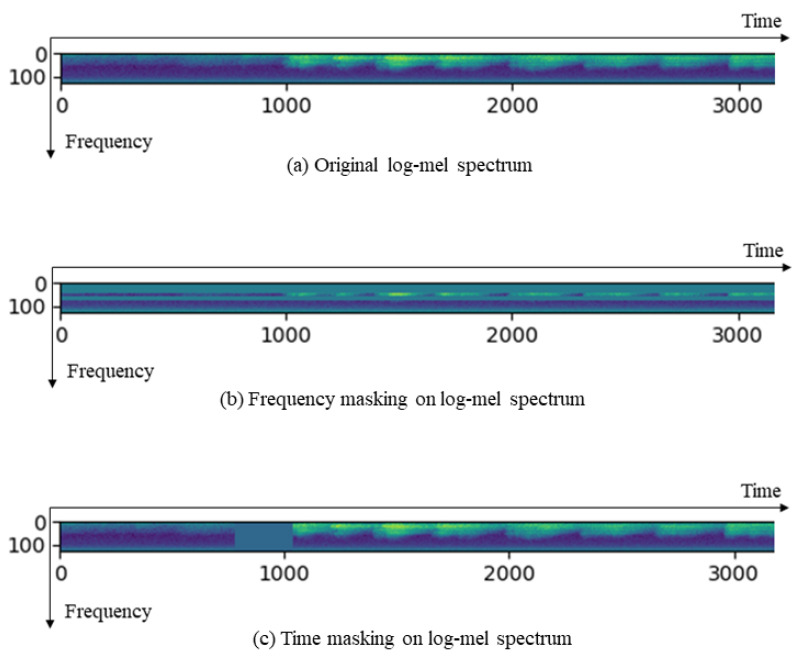
An example of frequency masking and time masking with SpecAugment. (**a**) is an original log-mel spectrum. (**b**) shows frequency masking on a log-mel spectrum, where two masks of different widths are applied on the frequency axis of the log-mel spectrum. (**c**) shows time masking on log-mel spectrum, where a mask is applied around the time coordinate of 1000 on the time axis of the log-mel spectrum.

**Figure 4 sensors-22-08015-f004:**
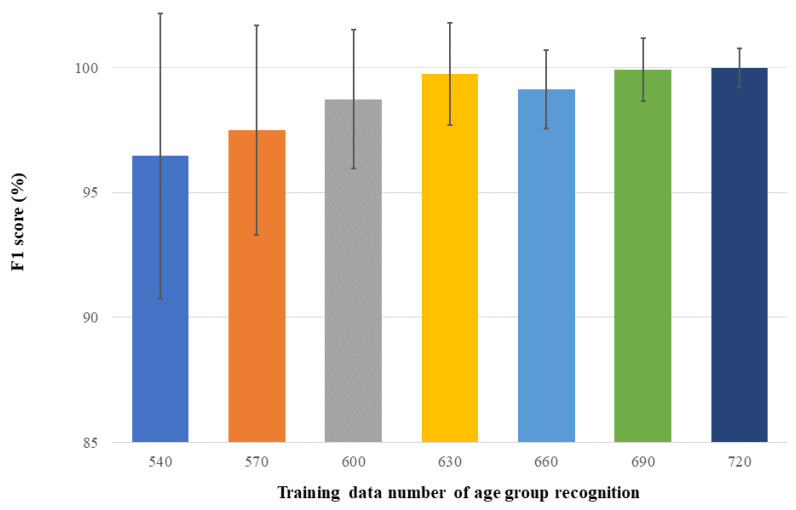
Average and standard deviation of F1 scores of the proposed method for age group recognition under different numbers of training data.

**Figure 5 sensors-22-08015-f005:**
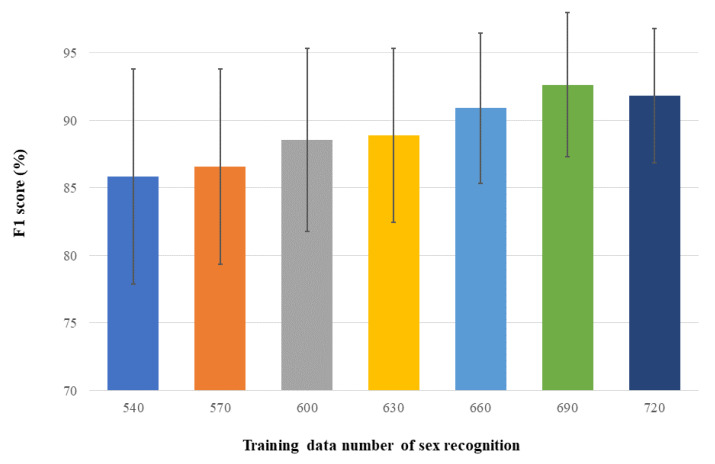
Average and standard deviation of the proposed method for sex recognition under different numbers of training data.

**Table 1 sensors-22-08015-t001:** The conditions and number of datasets for Experiment 1, Experiment 2, and Experiment 3.

Experiment	Training Data	Mutually Exclusive	Augmentation	Loss
1	540	No	No	Cross entropy
2A	540	Yes	No	Cross entropy
2B	540	Yes	Yes	Cross entropy
2C	540	Yes	Yes	Focal loss
3	540~720 (30 clips up)	Yes	Yes	Focal loss

**Table 2 sensors-22-08015-t002:** Training time for each experiment.

Experiment	Training Time
1	about 1.5 h
2A	about 1.5 h
2B	about 2 h
2C	about 2 h
3	about 13.5 h

**Table 3 sensors-22-08015-t003:** Performance of Experiment 1 and Experiment 2A-C of age groups recognition.

Mutually Exclusive	Augmentation	Loss	Precision	Recall	F1 Score
No	No	Cross entropy	99.35% ± 0.56%	97.23% ± 2.34%	98.28% ± 1.27%
Yes	No	Cross entropy	82.96% ± 10.43%	83.50% ± 15.03%	83.23% ± 13.21%
Yes	Yes	Cross entropy	92.58% ± 6.66%	85.43% ± 11.80%	88.86% ± 7.14%
Yes	Yes	Focal loss	97.43% ± 4.65%	95.51% ± 7.98%	96.46% ± 5.71%

**Table 4 sensors-22-08015-t004:** Performance of the proposed method for age groups recognition as the size of training data increases.

Training Data	F1 Score
540	96.46% ± 5.71%
570	97.50% ± 4.20%
600	98.73% ± 2.77%
630	99.75% ± 2.05%
660	99.13% ± 1.56%
690	99.91% ± 1.25%
720	99.98% ± 0.77%

**Table 5 sensors-22-08015-t005:** Performance of the Experiment 1 and Experiment 2A–C of sex recognition.

Mutually Exclusive	Augmentation	Loss	Precision	Recall	F1 Score
No	No	Cross entropy	96.75% ± 0.94%	92.24% ± 3.12%	94.44% ± 1.56%
Yes	No	Cross entropy	84.19% ± 12.84%	76.98% ± 19.42%	80.42% ± 15.64%
Yes	Yes	Cross entropy	86.63% ± 7.14%	77.43% ± 12.38%	81.77% ± 10.96%
Yes	Yes	Focal loss	91.93% ± 5.97%	80.52% ± 9.28%	85.85% ± 7.99%

**Table 6 sensors-22-08015-t006:** Performance of the proposed method for sex recognition as the size of training data increases.

Training Data	F1 Score
540	85.85% ± 7.99%
570	86.58% ± 7.23%
600	88.57% ± 6.79%
630	88.89% ± 6.44%
660	90.91% ± 5.58%
690	92.65% ± 5.34%
720	91.85% ± 4.98%

## Data Availability

All data generated or presented in this study are available upon request from the corresponding author. Furthermore, the models and code used during the study cannot be shared at this time as the data also form part of an ongoing study.

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
