# Peer review of "Automatic Recognition of Giant Panda Attributes from Their Vocalizations Based on Squeeze-and-Excitation Network"

_sensors, 2022, doi:10.3390/s22208015_

Round 1
Reviewer 1 Report
The work presented by the authors is interesting as another example of ML algorithms exploitation. I have just a pair of observations to share. First of all the presentation could be improved, for instance by adding schemes or items related to the processing of audio samples. Moreover the improvements added w.r.t. reference [21] have to be better specified, also in terms of data processing. The second aspect is related to the repetition of the same operations on data that have been already reported in [21], there is no need, and it is in fact not professional, to repeat the same phrases of the previous article.
Author Response
Point 1: Adding schemes or items related to the processing of audio samples.
Response 1:In this paper, we process the audio samples with three main operations before feeding them into the neural networks.
(i) Reducing the sampling rate. We converted the audio data from dual-channel to single-channel, and normalized the sampling rate of all recordings to 44,100Hz. According to Ref. [22], reducing the sampling rate to 44,100Hz can effectively reduce the time of data processing, while maintaining the Compact Disc (CD) quality of audio. (line 123~line 125)
(ii) Dividing call clips into 2-second segments, and applying zero-padding if the segments are shorter than two seconds. In audio signal processing, it is very common to pad the log-mel spectrum by filling zeros or copying a part of the log-mel spectrum to make its length consistent. However, according to Ref. [23], compared with padding via copying, zero-padding can reduce errors in the log-mel spectrum. (line 128~line 131)
(iii) Extracting acoustic features. According to Ref. [24], the main frequency component of speech is called formant, which carries the identification attributes of a sound, just like a personal ID card. We thus extracted the MFCC (Mel-frequency Cepstral Coefficients) features because it can contain formant information (see Ref. [25]). Please find more detail in Sections 2.1.2 and 2.1.3 in the revised manuscript.
Point 2: Moreover the improvements added w.r.t. reference [21] have to be better specified, also in terms of data processing.
Response 2:As an extended version of our previous conference paper in Ref. [21], we expanded the work mainly from three aspects. Please refer to Section 1.2 for more detail.
(i) More data. Vocalizations from sub-adult pandas were included in this paper, whereas the dataset in [21] contains only adult pandas and cubs aged under 100 days. (line 98~line 100)
(ii) More reasonable data organization. The data in this paper was carefully organized such that individuals do not overlap between training and testing data, while there is no such requirement in [21]. (line 100~line 102)
(iii) More attributes. We further showed in this paper that both age groups and sex of giant pandas can be automatically recognized from their vocalizations, which is implemented through the same network, while in [21] only the age groups were recognized. (line 102~line 104)
Point 3: The second aspect is related to the repetition of the same operations on data that have been already reported in [21], there is no need, and it is in fact not professional, to repeat the same phrases of the previous article.
Response 3:Thank the reviewer for the suggestion. This paper is an extension of our previously published conference paper in Ref. [21] (see Point 2). We include some of the contents in [21] for completeness such that readers can easily follow the work in this paper without going through Ref. [21] in detail.

Reviewer 2 Report
This research is significant as it attempts to determine the population of pandas from vocalizations. The method and results of suppressing the biased effect of the dataset for training are shown, and it seems that a certain level is secured as a paper.
However, I think the following points need to be clarified.
(1)From Figure 4, the authors mentioned that the more training data, the better the model becomes. What is the basis for this? Is it wrong to think that the effect is saturated after 630 data number? From the results of this graph alone, it seems difficult to draw conclusions like this paper. Please reinforce and provide the basis for your conclusion.
(2)This is the same question as (1). The authors concluded that the more training data, the better the F1 score is, but the score at the data number 720 is lower than that of 690. Please explain why there is no possibility that the data number 690 is a more appropriate value.
(3)Lines 268 to 269 describe the conditions for creating a training model, but the time required for training should also be described. Line 226 shows the training environment.
(4)Please clearly indicate the meaning of " ' " attached to recall', etc. in expressions (2), (3), and (4).
(5)To make it easier for the reader to understand, please add a table or figure summarizing the conditions and number of data sets for Experiments 1, 2, and 3.
Author Response
Point 1: From Figure 4, the authors mentioned that the more training data, the better the model becomes. What is the basis for this? Is it wrong to think that the effect is saturated after 630 data number? From the results of this graph alone, it seems difficult to draw conclusions like this paper. Please reinforce and provide the basis for your conclusion.
Response 1:Thank the reviewer for pointing out our unclear/irrigorous statements. In this revised paper, we have corrected the statements. To be specific, we use the average and standard deviation of accuracy (e.g., in terms of F1 scores) in ten-folder cross-validation evaluation experiments to measure the performance of different methods. A higher average F1 score and a smaller standard deviation are preferred. According to the results we obtained for age group recognition (refer to Fig. 4 and Table 4 in the revised paper), using more training data is beneficial to improving the performance of the recognition model. As the size of training data increases, the average accuracy in terms of F1 scores is in general enhanced. Although in our experiments the accuracy improvement becomes marginal and fluctuated as the number of training data reaches 630 or more, the standard deviation still keeps being reduced. Please refer to Section 3.2 for more detail.
Point 2: This is the same question as (1). The authors concluded that the more training data, the better the F1 score is, but the score at the data number 720 is lower than that of 690. Please explain why there is no possibility that the data number 690 is a more appropriate value.
Response 2:Sorry for our unclear/irrigorous statements. We have corrected the statements in this revised paper. Specifically, compared with age group recognition, similar observation can be made for sex recognition. As the number of training data increases, the improvement on average F1 gradually becomes slight and even fluctuates, whereas the standard deviation keeps reduced. Please refer to Section 3.3 for more detail.
Point 3: Lines 268 to 269 describe the conditions for creating a training model, but the time required for training should also be described. Line 226 shows the training environment.
Response 3:Thank the reviewer for the suggestion. In this revised paper, the training time for different experiments is reported in Table 2. Please refer to Section 2.3.
Point 4: Please clearly indicate the meaning of " ' " attached to recall', etc. in expressions (2), (3), and (4).
Response 4:This is in fact a typo, which has been corrected in this revised paper. We are very sorry for the confusion caused by our carelessness.
Point 5: To make it easier for the reader to understand, please add a table or figure summarizing the conditions and number of data sets for Experiments 1, 2, and 3.
Response 5:Table 1 is added in this revised paper to summarize the conditions and number of data sets in different experiments according to the constructive suggestion of the reviewer. Please refer to Section 2.3 for detail.

Round 2
Reviewer 2 Report
I have confirmed the changes and additions to the description regarding the pointed-out items.